# Multichannel One-Dimensional Data Augmentation with Generative Adversarial Network

**DOI:** 10.3390/s23187693

**Published:** 2023-09-06

**Authors:** David Ishak Kosasih, Byung-Gook Lee, Hyotaek Lim

**Affiliations:** Department of Computer Engineering, Dongseo University, Busan 47011, Republic of Korealbg@dongseo.ac.kr (B.-G.L.)

**Keywords:** data augmentation, one-dimensional data, generative adversarial network (GAN)

## Abstract

Data augmentation is one of the most important problems in deep learning. There have been many algorithms proposed to solve this problem, such as simple noise injection, the generative adversarial network (GAN), and diffusion models. However, to the best of our knowledge, these works mainly focused on computer vision-related tasks, and there have not been many proposed works for one-dimensional data. This paper proposes a GAN-based data augmentation for generating multichannel one-dimensional data given single-channel inputs. Our architecture consists of multiple discriminators that adapt deep convolution GAN (DCGAN) and patchGAN to extract the overall pattern of the multichannel generated data while also considering the local information of each channel. We conducted an experiment with website fingerprinting data. The result for the three channels’ data augmentation showed that our proposed model obtained FID scores of 0.005,0.017,0.051 for each channel, respectively, compared to 0.458,0.551,0.521 when using the vanilla GAN.

## 1. Introduction

Data augmentation is a problem for artificially generating new data from an existing dataset. It is widely used in many deep learning tasks to address data insufficiency and avoid overfitting, thus improving their performance [1,2]. Data augmentation techniques have solved problems in many areas, such as weather prediction [3], healthcare [4,5,6], speech emotion recognition [7], and sound classification [8]. Data augmentation techniques have also been used to attack and defend machine learning models [9,10,11,12].

Several techniques have been proposed. A survey on image data augmentation was conducted by [13]. These techniques include geometric transformation, color space augmentations, kernel filters, mixing images, random erasing, feature space augmentation, and deep learning-based methods. The most common deep learning-based image data augmentation method is the generative adversarial networks (GAN), originally proposed by [14]. Since then, several works have been proposed to improve GAN, such as those of [15,16,17]. Ref. [18] conducted a review of the GAN model for the video domain. Recently, ref. [19] proposed a novel method for augmenting image data, called diffusion probabilistic models. In evolutionary computation, ref. [8] proposed a crossover-based technique for data augmentation. Most of these works, however, centered on computer vision-related tasks. A few works on one-dimensional data include that of [8,20,21]. A comprehensive survey on time series data augmentation was also conducted by [22]. Most of these works used either manual data augmentation methods or a simple GAN.

In this paper, we proposed a multiple discriminator GAN to generate multichannel one-dimensional data. We use the first discriminator to analyze the general patterns across channels and the rest of the discriminator to extract the local information of each channel. We adapt DCGAN [15] for our first discriminator and patchGAN for the other discriminators [16]. Therefore, the main contributions of our proposed work are as follows:Adaptation of patchGAN and DCGAN for one-dimensional data augmentation.Multiple discriminators used to extract global and local information.Convolutional neural network (CNN) and residual-based generator for generating multichannel data given single channel inputs.Finally, we evaluate our proposed system with the website fingerprinting data proposed in [23].

The remainder of this paper is organized as follows: Section 2 summarizes the related work on data augmentation and GAN. We describe our general architecture in Section 3. We present our experimental results and discussion in Section 4. Finally, we conclude our work in Section 5.

## 2. Related Work

In this section, we review related GAN-based data augmentation methods.

Ref. [14] proposed a generative adversarial network (GAN) to generate image data. GAN consists of two models, namely, generator and discriminator. The former generates an image given a random noise, while the latter tries to tell if it is real or fake. Ref. [15] used convolutional layers to improve the generator. Ref. [16] proposed patchGAN to capture local information from the image. Unlike the original GAN discriminator, pathGAN predicts whether each patch is real or fake instead of predicting the entire image. Ref. [24] proposed a conditional GAN to consider additional information, such as labels. Ref. [25] proposed cycle-consistent adversarial networks for image-to-image translation without a paired example. Ref. [26] used a pre-trained model to extract several feature maps of the input and the generated output from different model layers. Each of the layer feature maps’ output pair is discriminated independently with different discriminators. Ref. [27] proposed the defect image generation method DG-GAN for detecting defects.

At the one-dimensional data level, ref. [20] solved the insufficient dataset problem for the fault diagnosis of electrical machines by adapting DCGAN. Although the proposed method could generate one-dimensional data that resembled the original input, it required separate training for each class. For example, given two classes, one needs to train to 1d DCGAN for each class. This is ineffective for a large number of classes, such as website fingerprinting. Ref. [8] used a cycle-consistent generative adversarial network (cycleGAN) to augment audio signals. However, instead of directly applying GAN on a one-dimensional audio signal, the authors transformed the data into a 2D representation using wavelet transform (DWT) before feeding it into cycleGAN. This increases computational overhead for data processing and may lead to information loss. Ref. [21] applied WGAN-GP for one-dimensional medical data augmentation to solve the problem of obtaining annotated medical data samples. Ref. [28] used DCGAN to generate artificial electroencephalogram (EEG) data. Similar to [8], they converted multichannel one-dimensional EEG data to images, which increases computational overhead and possibly leads to information loss. Moreover, the proposed methods in [8,20,21] cannot be applied directly to the multi-channels’ one-dimensional data augmentation task, which would otherwise help the classifier to extract more information across channels for each instance.

## 3. Methods

In this section, we will first introduce the general architecture of our proposed model. Next, we will show our generator and discriminator models. For simplicity, we will discuss the three channels throughout the rest of this paper.

### 3.1. General Architecture

Figure 1 illustrates our proposed model for generating three channels of 1D data given one channel input. We use one generator, one convolutional discriminator, and three patch discriminators. The first discriminator, $D_{1}$, receives three channels of one-dimensional data input from the generator and the real data (See blue discriminator in Figure 1). We use 1D convolutional layers in this discriminator to discriminate the input, similar to the architecture of DCGAN. Given three channels of 1D data input, each filter of the first convolutional layer will convolve through each input channel and aggregate the result to generate an output of *c* channels where *c* is the number of filters. This output will then be passed to several other convolutional layers before being flattened and passed through a fully connected layer to make the final decision of whether a given input is fake or real. The aggregate method of the convolutional layer implies that the final decision is derived from the combination of all three input channels; in other words, the decision is made based on the general pattern or the combined pattern of the three channels. This is certainly useful for extracting the combined pattern, but in addition to this, we also add another three discriminators ($D_{2},D_{3},D_{4}$) to determine whether each channel is real or fake. Each discriminator receives input from each channel, as illustrated in Figure 1. These three discriminators also use 1D convolutional layers, but instead of deciding whether the given input is real or fake, it generates several real or fake outputs where each output corresponds to the smallest area of the given input or is commonly referred to as a patch. This concept is known as patchGAN, introduced in [16]. The purpose of using patchGAN is to generate more precise local information, and thus, a better 1D data output. We explain this concept in Section 3.3.

Notice in Figure 1 that the input and the ground truth of the first channel come from the same one-dimensional data. This means that our generator tries to reconstruct the original data for the first channel and translates the original data to other one-dimensional data from the same class for the second and third channels.

### 3.2. Model Architecture


**Generator.** Figure 2 depicts the overall concept of our generator model. Unlike the common approach that rearranges the output such that it has the desired number of channels, we use convolutional filters to generate multichannel output. In this case, we use three filters to generate three channels of 1D data. We also use fully connected layers that receive input from the convolutional layer output to generate the final output. Finally, we use a skip connection that aggregates the original input with each channel of the final output to retain the original information.**PatchGAN discriminator.** We illustrate our patchGAN discriminator in Figure 3. We adapt patchGAN discriminator introduced in [16] to our discriminator for 1D data. Given an input of size *m*, we use four 1D convolutional layers to generate a final representation of size *n*, where n<m. In this way, each output value corresponds to a certain local area of the input, known as a patch. If the input comes from the real data, we generate a label vector of those with a size of *n*. Otherwise, we generate a vector with a size of *n* with each element to zero. Finally, we use a standard binary cross-entropy objective function to calculate the loss between the output and the label vector; thus, the discriminator determines whether each patch is real or fake.


### 3.3. Objective Function

Our discriminator loss is formulated by:(1)$$\begin{array}{c} L_{D}^{i}=log\left(D_{i}\left(x\right)\right)+log\left(1-D_{i}\left(G\left(x\right)\right)\right) \end{array}$$
where *i* is the index of the discriminator. There is no difference between calculating the loss of $D_{1}$ and the patch discriminators ($D_{2},D_{3},D_{4}$) since we can use the standard binary cross entropy loss for the patchGAN, as explained in Section.

Our generator loss is formulated by:(2)$$\begin{array}{c} L_{G}=log(D_{1}\left(G\left(x\right)\right)+\frac{1}{n}\sum_{i=2}^{n+1}log(D_{i}\left(G\left(x\right)\right) \end{array}$$
where *n* is the number of patch discriminators. In Equation (Equation 2), we sum the loss of the discriminator $D_{1}$ with the average loss of the patch discriminators’ losses.

## 4. Experiments

In this section, we compare our method with vanilla GAN and Wasserstein GAN (WGAN) [29] for one-dimensional data that are widely used. Typically, this model is fully connected for both generator and discriminator.

### 4.1. Dataset and Setup

We evaluate our proposed work with website fingerprinting data, called RND-WWW, proposed by [23]. This dataset is widely used by the Tor research community to analyze users’ access through the Tor browser. RND-WWW contains TCP traffic data to more than 120.000 unique webpages with up to 40 times access to each web page. For our experiment, we only used 19 websites. Each website contains ±50 sub-pages with 15 having access to each page; therefore, we have a total of 14.250 instances. We split up these instances for the training and testing of our GAN; more precisely, we used 14 instances for training and one instance for testing. Figure 4 illustrates the structure of our dataset and how we split the training and testing data.

Each TCP traffic contains the number of incoming packets, the number of outgoing packets, incoming packet sizes, and outgoing packet sizes. We do not apply any data preprocessing beyond what is proposed in [23] and data scaling. The data preprocessing consists of two steps: cumulative representation and sampling of the piecewise linear interpolation to obtain the same data length for each instance. We then normalize the data to 1 and 0. The normalization method is formally defined in Equation (Equation 4):(3)$$\begin{array}{c} \vec{p_{i}^{\prime }}=\frac{\vec{p_{i}}-min\left(\vec{p_{i}}\right)}{max\left(\vec{p_{i}}\right)-min\left(\vec{p_{i}}\right)} \end{array}$$
where $p_{i}$ denotes the data point.

We used the Fréchet inception distance (FID) score proposed by [29] to measure the performance of our proposed model compared to vanilla GAN for one-dimensional data. FID score is a metric commonly used to measure the quality of the generated image. We adapted this metric to measure the performance of our one-dimensional generated data. A lower FID score implies a better quality of the generated data.

We ran all of our experiments for 15 epochs. The learning rates were set to 3e−3 for all discriminators and 8e−3 for the generator. A lower learning rate for the generator is known to yield a better result. We used stochastic gradient descent as our optimizer since the model may converge much faster, especially for large datasets. Our generator used one 1D convolutional layer and four fully connected layers with SELU as the activation function. We used three 1D convolutional layers for the main discriminators, followed by one fully connected layer. Four 1D convolutional layers were used for all patch discriminators without any fully connected layer. All our discriminators used ELU for the hidden layers’ activation function and sigmoid for the output layer. We used SELU for the generator because it may be important for the generator to always standardize the output.

### 4.2. Results

As aforementioned, we experimented with 19 websites. Figure 5 shows the comparison between the generated data using vanilla GAN and the original data for two specific websites: BBC and Facebook. The model could better generate BBC data; however, the overall results did not resemble the original data.

Based on the result with vanilla GAN, we hypothesized that the poor performance was due to the discriminator’s inadequacy to extract patterns from one-dimensional data. To address this issue, we tried to modify the discriminator by adding three 1D CNN layers, similar to the DCGAN architecture. However, the difference was that instead of using convolutional transpose layers, we still used a fully connected model for the generator. Since our objective was to generate a multichannel one-dimensional data output based on a single channel input, the input and output have the same length. Therefore, incorporating convolutional transpose layers into the generator architecture was not feasible.

Figure 6 shows the result with a modified discriminator. The results for the BBC and Facebook data implied that a fully connected generator could generate multichannel one-dimensional data that starts to resemble the original data when it competes with the 1D CNN discriminator. In other words, our results indicated that by only enhancing the discriminator, we can make significant improvements to the performance of our GAN model.

In addition, we also compared our method with WGAN. The model we used is still the same, namely a fully connected generator and a 1D CNN discriminator, but as proposed in the original paper [29], we added a gradient penalty to the loss function to stabilize the training. As can be seen in Figure 7, this method can generate Facebook data fairly comparable to those of vanilla GAN with a modified discriminator; however, it performed worse for the BBC data.

Finally, we used a 1D CNN for both the generator and the discriminators. We also integrated patchGAN into our system, as explained in Section 3.3. The results in Figure 8 for BBC and Facebook indicated that our model could generate multichannel one-dimensional data that closely resemble the original data.

We compared the FID score of each channel between vanilla GAN, vanilla GAN with a modified discriminator, WGAN with a modified discriminator, and ours in Table 1. Our model significantly outperformed the vanilla GAN and performed better than the vanilla and CNN discriminator and the WGAN and CNN discriminator, especially for the first channel. This indicates that our model performed better in reconstructing the original data due to the ability to extract the local information. However, for the third channel, WGAN performed slightly better than ours.

Additionally, we provide a visual comparison of each channel for the generated BBC and Facebook data in Figure 9. Figure 9 indicates that the third channels of both Facebook and BBC are slightly noisier. This result agrees with the FID score assessment in Table 1, where the third channel’s FID score of the entire dataset was slightly higher than that of the first and third channels.

### 4.3. Discussion

The total number of parameters of our multiple discriminator is defined as follows:(4)$$\begin{array}{c} p_{total}=p_{D_{1}}+\sum_{i=2}^{i=n+1}p_{D_{i}} \end{array}$$
where *n* is the number of path discriminators, which corresponds to the number of channels. $p_{D_{i}}$ is the number of parameters of discriminator $D_{i}$, while $p_{total}$ is the total number of parameters. In this experiment, the first discriminator $D_{1}$ had 50.945 parameters, while each of the patch discriminators ($D_{2},D_{3},D_{4}$) used 39.745 parameters. Therefore, the total number of parameters $p_{total}$ is 50.945+3x39.745=170.180 as we used three channels. Compared to the models for simple computer vision tasks, such as classifying CIFAR-10, the number of parameters of each patch discriminator is significantly lower. If we consider 3.000.000 parameters to be the average number of parameters, then we can use 74 patch discriminators to reach the total of approximately 3.000.000 parameters for all discriminators, which means generating 74 channels of one-dimensional data given one channel’s one-dimensional data input.

We also observed that training our multiple discriminator GAN for generating three channels of one-dimensional data took approximately 5 min 2 s for each epoch. On the other hand, the vanilla GAN with a CNN discriminator (the adaptation of DCGAN) took 1 min 20 s to accomplish the same task. Since, in most cases, 15 epochs are enough to train our model, there is no significant time difference in the total training time. We use NVIDIA GeForce RTX 2080 to run all our experiments.

During the inference phase, we ignore all the discriminators and only use one generator. Therefore, our system does not introduce additional complexity and computational overhead during inference.

The results indicate that vanilla GAN with a fully connected discriminator and generator may work for simple one-dimensional data. However, for a more complicated and large dataset, it is unable to properly generate one-dimensional data that mimic the original data. Moreover, we showed that using 1D convolutional layers in the discriminator can significantly improve our GAN as it now has the ability to extract the pattern of the data; thus, unless the fully connected generator generates an output with the original pattern, the discriminator would deem it fake. Adding WGAN to this method did not significantly improve the performance. Therefore, instead of adding WGAN, we used a 1D convolutional layer in the generator to generate the multichannel data before feeding it to the fully connected layers and added several more discriminators, which adapted patchGAN for capturing the local information of each channel to enhance the overall performance. The experimental result showed that our method is significantly better than the vanilla GAN and the vanilla GAN with 1D convolution layers for the discriminator, especially for the first channel, which is the reconstruction of the input. It also outperformed WGAN with a 1D convolutional layer for the discriminator except for in the third channel.

Although our proposed work worked well for website fingerprinting data, a more extensive experiment with various datasets is still needed to investigate the side effect of using multiple discriminators. Future research may focus on analyzing the potential issue of our proposed multiple discriminators for various one-dimensional data, exploring various models for the generator and discriminators, and, more importantly, understanding how the models make decisions.

Lastly, our proposed work may apply to any one-dimensional data classification task. Similar to computer vision tasks, training a model with multi-channel data generally provides more information for the model, especially when we use convolutional layers. However, in the case of one-dimensional data, most of the data is one-channel data. We may combine these data to create multi-channel data during training, but for inference, combining the data to create the same number of channels as the training data may not always be possible. Simply duplicating the input data causes the model to lose the general information across channels. Therefore, in this paper, we attempted to provide a data augmentation solution with a multiple-discriminator GAN that adapts DCGAN and PatchGAN to generate multi-channel data that mimic its original input, thus alleviating the problem of multi-channel data for inference.

## 5. Conclusions

This paper proposed a GAN-based model to generate multichannel one-dimensional data given single-channel inputs. We proposed the use of multiple discriminators. We also adapted DCGAN and patchGAN to capture the general and local pattern of the generated data.

We evaluated our model on a website fingerprinting dataset. The result showed that our proposed model significantly outperformed vanilla GAN with a fully connected generator and discriminator that is commonly used for generating one-dimensional data. We also showed that only adding 1D CNN layers to the discriminator can significantly improve the performance of the generator.

## Figures and Tables

**Figure 1 sensors-23-07693-f001:**
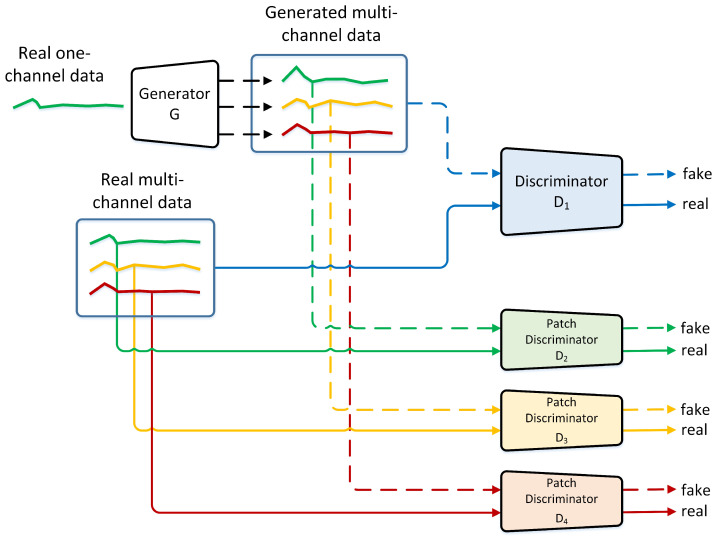
Overview of our proposed model. We use one discriminator to analyze the overall pattern and three discriminators to analyze each channel.

**Figure 2 sensors-23-07693-f002:**
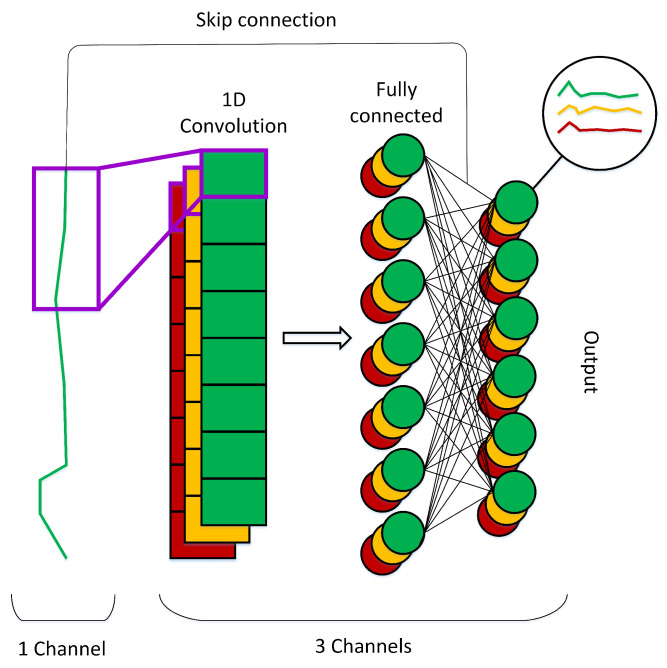
The generative model: we use a 1D convolutional layer to generate three channels of one-dimensional data, we feed the output of the convolutional layer to the fully connected layer to generate the final output, and we also use a skip connection between the input and the final output.

**Figure 3 sensors-23-07693-f003:**
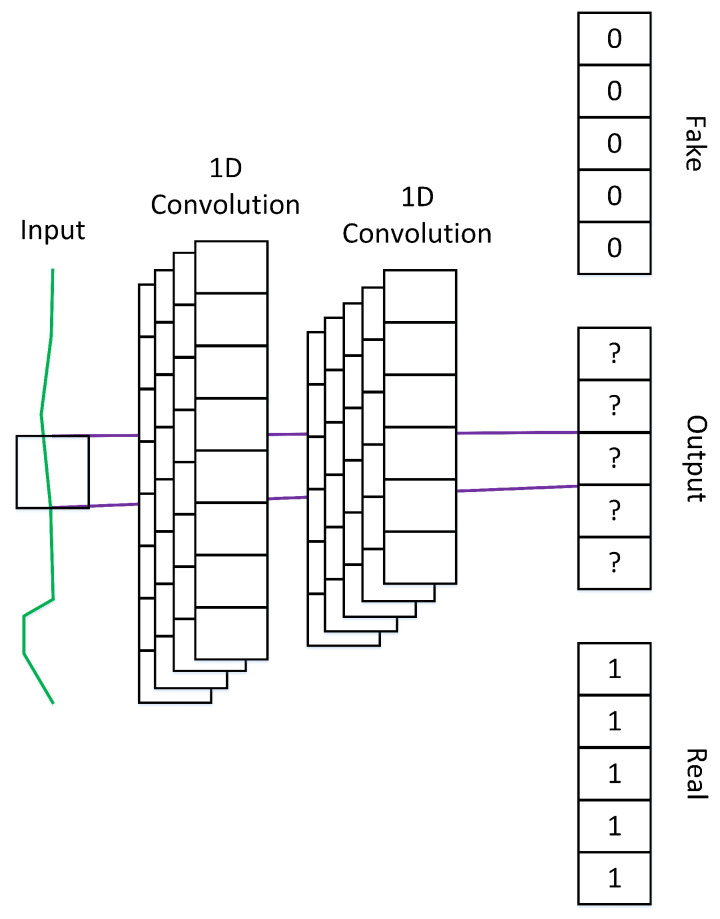
PatchGAN for each channel discriminator. Similar to the original patchGAN, the discriminator tried to predict if each patch is fake or real instead of predicting the whole input.

**Figure 4 sensors-23-07693-f004:**
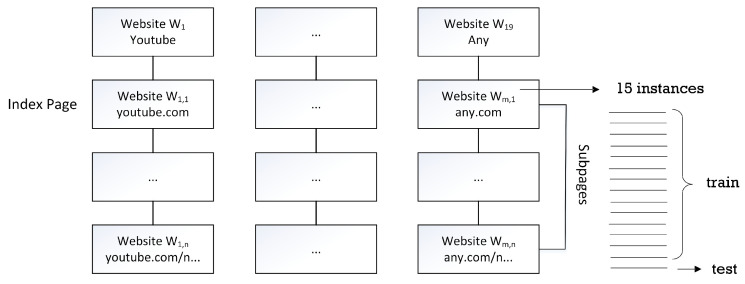
The structure of our dataset and the train–test split.

**Figure 5 sensors-23-07693-f005:**
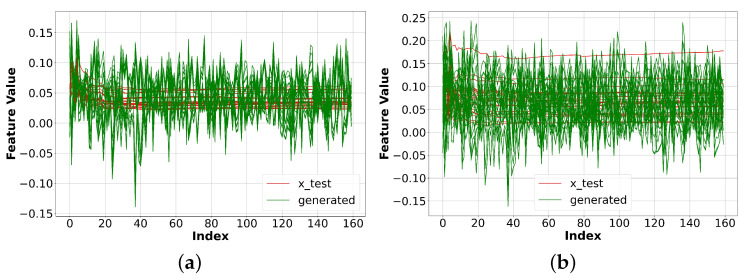
Experiment results with vanilla GAN. The red line represents the original data, while the green line represents the generated data. The results show that the generated data do not resemble the original data. (**a**) Original and generated BBC data. (**b**) Original and generated Facebook data.

**Figure 6 sensors-23-07693-f006:**
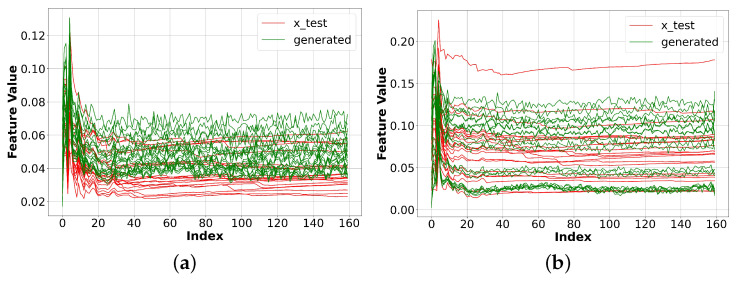
Experiment results with vanilla GAN and a modified discriminator. When we use 1D CNN for the discriminator, the generated data starts to resemble the original data. (**a**) Original and generated BBC data. (**b**) Original and generated Facebook data.

**Figure 7 sensors-23-07693-f007:**
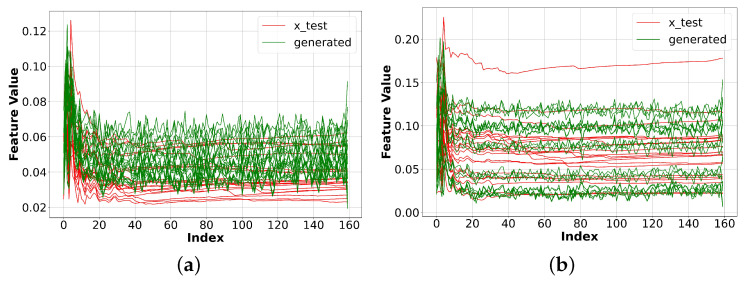
Experiment results with WGAN and a modified discriminator. The result for BBC was slightly worse than that for vanilla GAN with a modified discriminator. (**a**) Original and generated BBC data. (**b**) Original and generated Facebook data.

**Figure 8 sensors-23-07693-f008:**
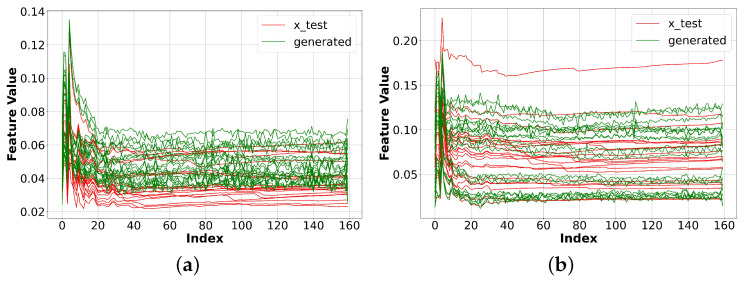
Experiment results with our proposed model. The results show that the generated data slightly more closely resembled the original data compared to CNN GAN, and far outperformed the vanilla GAN. (**a**) Original and generated BBC data. (**b**) Original and generated Facebook data.

**Figure 9 sensors-23-07693-f009:**
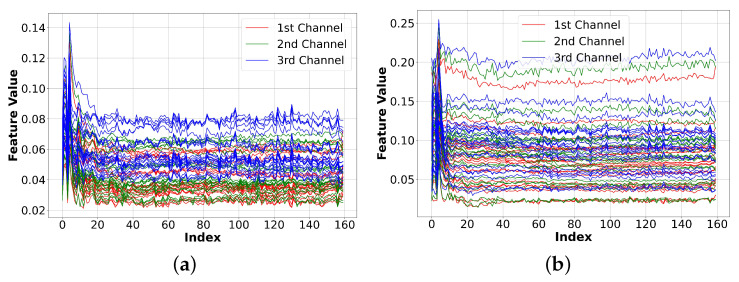
Visual comparison of each channel for the generated BBC and Facebook data. The results show that the 3rd channel is slightly more noisy than the other channel. (**a**) Generated BBC data. (**b**) Generated Facebook data.

**Table 1 sensors-23-07693-t001:** FID score for each channel. Our method had significantly lower FID scores than those from vanilla GAN for one-dimensional data.

Method	1st Channel FID Score	2nd Channel FID Score	3rd Channel FID Score
Vanilla	0.4583	0.5510	0.5212
Vanilla & CNN discriminator	0.0270	0.0361	0.0588
WGAN & CNN discriminator	0.0489	0.0331	**0.0392**
Ours all	**0.0053**	**0.0167**	0.0513

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
