# Peer review of "Multichannel One-Dimensional Data Augmentation with Generative Adversarial Network"

_sensors, 2023, doi:10.3390/s23187693_

Round 1

Reviewer 1 Report

This paper introduces a GAN-based approach for data augmentation of multi-channel one-dim data. It seems like this paper proposes an architecture for generating three channels of 1-dim data given one channel input using a combination of GAN and PatchGAN techniques. Here are some observations and potential issues:

1- The proposed architecture involves multiple discriminators (one main discriminator and three patch discriminators) along with the generator. While this approach may or may not provide more local information and potentially enhance the quality of generated data, it also introduces additional complexity and computational overhead, especially when training the model. Training multiple discriminators can be more challenging, and it might require careful tuning of hyperparameters to ensure stability. In addition, understanding what the model has learned and how it is making decisions might become difficult due to the increased complexity. Some discussion regarding these issues is necessary.

2- GAN training can be notoriously difficult to stabilize, and using multiple discriminators might exacerbate this issue. It is extremely hard to ensure that the model converges and produces meaningful results and that might require extensive experimentation and adjustments in the architecture and training process

3- The direct application of PatchGAN to 1-dim data may not be as straightforward. The effectiveness of PatchGAN for 1-dim data must be thoroughly justified. 

4- How can one ensure optimal performance with all these additional layers? this might require a significant amount of experimentation and computational resources.

5- Lack of enough experimental results. Only one data set is considered and it is only considered to vanilla GAN. A lot more investigation and comparison to other models are required. 

it is not hard to follow. Just minor changes in the way texts are presented would help the readers to understand the paper better. Having that said, this paper sounds very ambitious and lacks details and it is very hard to understand the details of the paper. 

Reviewer 2 Report

The paper presents an interesting and potentially valuable contribution to the field of data augmentation for one-dimensional data using GANs. Addressing the above-mentioned points will significantly improve the manuscript's quality and ensure its relevance to the research community.

- The introduction provides a clear context for the problem of data augmentation in one-dimensional data, and the motivation for proposing a GAN-based approach is well-founded. However, it would be helpful to provide a more comprehensive literature review to highlight the limitations of existing data augmentation techniques for one-dimensional data. This will strengthen the paper's contribution and provide a clearer justification for the proposed GAN-based approach.

-While the authors mention that most existing data augmentation works are focused on computer vision tasks, it would be beneficial to include a more detailed comparison with the few works that have addressed data augmentation for one-dimensional data. This will help readers understand the novelty and significance of the proposed model in the context of the existing literature.

- The architecture of the proposed GAN-based data augmentation model is adequately described. However, the authors should consider providing more implementation details and hyperparameters used in training the model. Additionally, explain why the specific number of discriminators and layers were chosen for the architecture. This information will be valuable for reproducibility and understanding the model's behavior.

-The use of website fingerprinting data for experimentation is appropriate for evaluating the proposed approach's performance. However, the authors should provide more information on the dataset used, data preprocessing, and any potential biases that could affect the results. Additionally, include a brief explanation of the evaluation metrics used, not only FID scores but also other relevant metrics for assessing data augmentation quality.

-The reported FID scores for the proposed model and vanilla GAN are promising. However, to strengthen the analysis, the authors should consider conducting a more comprehensive comparison with other state-of-the-art data augmentation techniques for one-dimensional data. This will help demonstrate the superiority of the proposed method over existing approaches.

- The discussion section should elaborate on the key findings, limitations of the proposed model, and potential directions for future research. Additionally, highlight the practical implications and applications of the proposed GAN-based data augmentation approach beyond website fingerprinting data.

-Including visualizations or examples of the generated multichannel data would significantly enhance the clarity of the paper and help readers grasp the quality of the augmentation. Provide qualitative assessments and examples of the generated data for different channels.

- The overall clarity and organization of the paper are reasonable. However, some sentences may need revision for better readability and grammar. Proofread the manuscript thoroughly to avoid any language-related issues.

Round 2

Reviewer 1 Report

I want to express my gratitude to the authors for addressing some of my concerns. While some of my concerns remain unaddressed, I appreciate the effort made. The quantity of parameters presented in the review lacks coherence and necessitates further elucidation. Moreover, I maintain the view that certain elucidations offered in the review should be seamlessly integrated into the manuscript. This integration is crucial for a comprehensive understanding of the model's strengths and limitations. 

Additionally, I strongly recommend that the authors undertake a comparative analysis of their model against other contemporary concepts. The models highlighted in the paper do not adequately substantiate the superiority of the proposed approach. A more comprehensive account of the experimental methodology and intricate details is imperative for a robust presentation. This will undoubtedly enrich the reader's comprehension of the research and its implications.

It is written fine, however some minor changes may be required.

Reviewer 2 Report

Accepted

Minor editing of English language required

Author Response

Thank you for the reviewer effort to improve our paper.